# Leukocyte mitochondrial DNA copy number is a potential non-invasive biomarker for psoriasis

**Materah Salem Alwehaidah**[1]*, **Suad AlFadhli**[1], **Ghada Al-Kafaji**[2]

**1** Faculty of Allied Health, Department of Medical Laboratory, Kuwait University, State of Kuwait,
**2** Department of Molecular Medicine and Al-Jawhara Centre for Molecular Medicine, Genetics, and Inherited Disorders, College of Medicine and Medical Sciences, Arabian Gulf University, Manama, Kingdom of Bahrain

* matrasalem2016@gmail.com, matra.alweheda@ku.edu.kw

## Abstract

Abnormalities in the mitochondria have been linked to psoriasis, a chronic immune-mediated inflammatory skin disease. The mitochondrial DNA (mtDNA) is present in thousands of copies per cell and altered mtDNA copy number (mtDNA-CN), a common indicator of mitochondrial function, has been proposed as a biomarker for several diseases including autoimmune diseases. In this case–control study, we investigated whether the mtDNA-CN is related to psoriasis, correlates with the disease duration and severity, and can serve as a disease biomarker. Relative mtDNA-CN as compared with nuclear DNA was measured by a quantitative real-time polymerase chain reaction in peripheral blood buffy coat samples from 56 patients with psoriasis and 44 healthy controls. The receiver operating characteristic (ROC) curve analysis was performed to evaluate the value of mtDNA-CN as a biomarker. We found that the mtDNA-CN was significantly decreased in patients with psoriasis compared to healthy controls (93.6±5.3 vs. 205±71; P = 0.04). Sub-group analyses with stratification of patients based on disease duration under or over 10 years and disease severity indicated that the mtDNA-CN was significantly lower in patients with longer disease duration (74±4.3 in disease duration >10 years vs. 79±8.3 in disease duration <10 years, P = 0.009), and higher disease severity (72±4.3 in moderate-to-severe index vs. 88.3 ± 6 in mild index, P = 0.017). Moreover, the mtDNA-CN was negatively correlated with the disease duration and disease severity (r = -0.36, P = 0.006; r = -0.41, P = 0.003 respectively). The ROC analysis of mtDNA-CN showed an area under the curve (AUC) of 0.84 (95% confidence interval: 0.69–0.98; P = 0.002) for differentiating patients from healthy controls. Our study suggests that low mtDNA-CN may be an early abnormality in psoriasis and associates with the disease progression. Our study also suggests that mtDNA-CN may be a novel blood-based biomarker for the early detection of psoriasis.

## Introduction

Psoriasis is a chronic immune-mediated inflammatory skin disease, affecting approximately 2–3% of the worldwide population [1]. Clinical manifestation is characterized by the

---

**Data Availability Statement:** Data cannot be shared publicly because of data contain potentially identifying or sensitive patient information. Data are available from the Health Science Centre Ethics Committee at Kuwait University (hsc.

---

ethicalcommitee@ku.edu.kw) for researchers who
meet the criteria for access to confidential data.

**Funding:** The author(s) received no specific
funding for this work.

**Competing interests:** The authors have declared
that no competing interests exist.

appearance of erythematosquamous papules or plaques of various sizes, which are typically symmetrically distributed over the knees, elbows, genital area, scalp, and body [2–4]. Psoriasis is associated with several comorbidities such as type 2 diabetes millets, cardiovascular disease, and hypertension [3,5]. The appearance of comorbidities correlates with the severity of clinical presentation, which is assessed by the Psoriasis Area severity index (PASI) and is usually increases with age or disease duration [5,6]. Although the etiology of psoriasis remains ambiguous, the disease is considered multifactorial, involving an interplay between genetics and environmental factors [6–8]. At the genetic level, studies have revealed the association of genetic variations in the human leukocyte antigen (*HLA*) genes and non-*HLA* genes with the risk of psoriasis [9]. Emerging evidence suggests that the mitochondria are important regulators of keratinocyte development and differentiation [10], which are abnormally regulated in psoriasis [11]. The primary function of mitochondria is energy production in the form of adenosine triphosphate (ATP) through the process of oxidative phosphorylation (OXPHOS). During this process, reactive oxygen species (ROS) are generated as by-products of oxygen metabolism. The mitochondria are also involved in other essential cell functions such as regulation of calcium homeostasis, insulin secretion, innate immune and inflammatory responses, and apoptosis [12].

Each mitochondrion carries several copies of its own DNA (mtDNA), a circular double-stranded DNA molecule of about 16.568 kb. Human mtDNA contains 37 genes encoding 13 protein subunits of the electron transport chain (ETC) complexes that are involved in the OXPHOS, as well as 22 tRNAs and 2 rRNAs, all of which are important for normal mitochondrial function [13]. The number of mtDNA copies is highly dynamic and regulated in a cell-specific manner by mechanisms that are not fully understood [14]. Estimation of mtDNA copy number (mtDNA-CN) is often determined by the ratio of mtDNA to nuclear DNA (nDNA), which indicates the number of mitochondrial genomes per cell. Since mtDNA encodes most of the subunit genes of the OXPHOS system, mtDNA-CN is associated with mitochondrial gene stability and mitochondrial biogenesis and is considered as a surrogate measure of mitochondrial function [14].

The mtDNA is particularly vulnerable to oxidative stress because of inadequate DNA repair pathways, absence of protective histone, and high ROS exposure [15]. Oxidative damage to mtDNA can result in mutations and alterations in mtDNA replication and/or transcription efficiency and mtDNA-CN, which may subsequently lead to a decline in mitochondrial function with more ROS production [16–19]. Indeed, impaired mitochondrial function and increased oxidative stress have been implicated in the aging process and various human diseases including psoriasis [17–21].

Evidence also indicates an essential role of the interleukin (IL)-23/IL-17 axis and dendritic cell-T cell crosstalk in the development of skin inflammation through mitochondrial ROS [22] suggesting a link between increased oxidative stress and inflammation-induced mitochondrial impairment [23]. Different other mitochondrial abnormalities and mtDNA defects have been reported in psoriasis and other skin diseases [24]. For instance, common mtDNA single nucleotide polymorphisms (SNPs) which can be broadly categorized into mitochondrial haplogroup have been linked to psoriasis. In this context, an association between mitochondrial European haplogroup U and elevated IgE has been reported in children with atopic dermatitis [25]. In our previous studies, we found that haplogroup M can increase the risk of psoriasis in an Arab population [26] and that mtDNA variations play a role in the pathogenesis of psoriasis [27]. Moreover, changes in the mtDNA content and expression of mitochondrial regulatory proteins have been described in psoriasis and are implicated in the pathogenesis of the disease [28]. Alterations in the mtDNA-CN have been also described in several diseases in which oxidative stress plays a significant role. Specifically, decreased mtDNA-CN in peripheral blood

was reported in a number of autoimmune diseases such as rheumatoid arthritis and multiple sclerosis [29,30]. Decreased peripheral blood mtDNA-CN was also associated with cancer [31], type 2 diabetes [32], metabolic syndrome [33] and correlated with stroke [34], and the severity of coronary heart disease [35].

In the present study we aimed to 1) investigate changes in the mtDNA-CN in patients with psoriasis and healthy controls, 2) determine if mtDNA-CN is correlated with the disease duration or disease severity, and 3) determine the value of mtDNA-CN as a biomarker for psoriasis. Given that mtDNA-CN is an indicator of mitochondrial function and because mitochondrial dysfunction is involved in the pathogenesis of psoriasis, we hypothesized that the mtDNA-CN would be decreased in psoriasis patients and can be utilized as a non-invasive biomarker for psoriasis.

## Methods

### Subject

A total of 100 subjects were enrolled in this study, including 56 patients with psoriasis and 44 unrelated healthy control individuals. Patients were recruited from the Suaid Al-Subah Dermatology Centre in the State of Kuwait and were diagnosed clinically based on the presence of typical erythematous scaly patches and papules. The severity of the disease was determined by Psoriasis Area Severity Index (PASI) within a score range of 0–72 [36]. Mild psoriasis patients had a PASI score of up to 10, moderate to severe psoriasis patients had a PASI score of 10–20, and severe psoriasis patients had a PASI score above 20. Healthy control subjects were free from inflammatory dermatoses or autoimmune diseases and were recruited from Central Blood Bank, State of Kuwait. Clinical data of patients such as disease duration (time from onset of disease to blood collection), PASI score, and type of medication (topical or systemic treatment) were collected from their medical records. Demographic data including age and gender were also reported for patients and controls. Written informed consent was obtained from each participant under a protocol approved by the Health Science Centre Ethics Committee at Kuwait University and the Health and Medical Research Committee in the Ministry of Health in Kuwait (No. 2016/496).

### Extraction of genomic DNA

From each participant, 5 ml blood was collected in ethylenediaminetetraacetic acid (EDTA) tubes. Blood samples were promptly centrifuged at 1000 g for 15 min to separate the buffy coat for genomic DNA extraction using the QIAamp DNA Mini Kit (Qiagen, Germany) according to the manufacturer's instructions. In brief, 200 μl of buffy coat was mixed with 20 μl of protease. Lysis buffer (200 μl) was added, and the mixture was incubated at 56°C for 10 min. This was followed by centrifugation at 20,000 g for 1 min. Then 200 μl of absolute ethanol was added and the mixture was centrifuged at 6000 g for 1 min. Washing buffer (500 μl) was added twice and centrifuged at 6000 g for 1 min then at 20,000 g for 3 min. 200 μl of elution buffer was added to elute the genomic DNA in clean tubes, then incubated at room temperature for 1 min, and centrifugated at 6000 g for 1 min. Each DNA sample was checked for purity using a NanoDrop 1000 system (Thermo Fisher Scientific) and for concentration using a Qubit 3.0 Fluorometer (Thermo Fisher Scientific).

### Determination of mtDNA copy number

The mtDNA copy number (mtDNA-CN) per nuclear genome was determined by quantitative real-time PCR (qPCR) using Power SYBR® Green PCR Master Mix (Applied Biosystems;

Thermo Fisher Scientific, Inc.). NADH dehydrogenase subunit 2 *(ND2)* gene was used as the target sequence for the determination of mtDNA (Forward primer 5'- `CAC AGA AGC TGC CAT CAA GTA`-3' and reverse primer 5'- `CCG GAG AGT ATA TTG TTG AAG AG`-3'). Beta-2-macroglobulin (*β2M*) was used as an internal reference gene (Forward primer 5'-`CCA GCA GAG AAT GGA AAG TCA A`-3' and reverse primer 5'-`TCT CTC TCC ATT CTT CAG TAA GTC AAC T`-3'). Genomic DNA (10 ng) was mixed with 1X Power SYBR® Green PCR Master Mix, forward and revers primers (50 nM each), and nuclease-free water to a final volume of 10 μl. qPCR was performed with a 7900HT real-time PCR system (Applied Biosystems; Thermo Fisher Scientific, Inc.) under the following conditions: denaturation at 95˚C for 10 minutes followed by 40 cycles of 10s at 95˚C, 30s at 60˚C, and 30s at 72˚C. Experiments were done in duplicate and non-template control (with omitted DNA) was included in each run. Relative quantitation of mtDNA-CN was obtained from the Ct values of *ND2* as the target gene, and the Ct values of *β2M* as the reference gene. ΔCt (CtND2—Ct$_{β2M}$) values were then obtained for cases and controls and the relative mtDNA-CN was calculated using the $2^{-\Delta\Delta ct}$ method.

## Statistical analysis

Statistical analysis was done with the Statistical Package for the Social Sciences (SPSS, version 20.0; IBM Corp., Armonk, NY, USA). First the normal distribution of the data was assessed by the Kolmogorov-Smirnov test. The results showed a P value of 0.2, confirming the normal distribution of data. Accordingly, the equivalent non-parametric Wilcoxon paired t-test, and Mann–Whitney test were used to compare the variables between cases and controls. Spearman correlation analysis was performed to determine the correlation between mtDNA-CN and clinical variables. The receiver operating characteristic (ROC) curve analysis was performed to evaluate the value of mtDNA-CN as a biomarker. In this analysis, the area under the ROC curve (AUC) and 95% confidence interval (95% CI) were obtained. A P<0.05 was considered statistically significant.

## Results

### Characteristics of psoriasis patients and healthy controls

A total of 56 patients with psoriasis and 44 healthy controls were included in this study. The demographics and clinical characteristics of patients and controls are shown in Table 1. Data are presented as number, percentage (%) or mean ± standard deviation (SD). The patient group comprised 28 males and 28 females with a group mean age of 39 ± 5. The healthy control group comprised 19 males and 25 females with a group mean age of 30 ± 7. No significant difference was observed in the gender distribution between patients and controls (P = 0.47), while a significant difference was found in the mean age between the two groups (P = 0.001). The disease duration in the patient group ranged from 1–40 years, with a mean of 16 ± 1.7. The mean of PASI score for disease severity was 16 ± 2. Twenty-three patients had mild psoriasis (PASI score: 5 ± 0.5), and 33 patients had moderate to severe psoriasis (PASI score: 24 ± 2). The patients were undergoing the following treatment: Topical treatment with corticosteroids cream (n = 20), or systemic treatment with Adalimumab, Etanercept, Methotrexate, Ixekizumab, Secukinumab and Ustekinumab (n = 36).

### mtDNA-CN is decreased in psoriasis patients

The mtDNA-CN was quantified by real-time qPCR as the DNA ratio (mtDNA/nDNA) between a target mitochondrial gene (*ND2*) and a reference nuclear gene (*β2M*) in peripheral

**Table 1. Demographics and clinical characteristics of psoriasis patients and healthy controls.**

| Characteristics | Psoriasis | Controls | P value |
|---|---|---|---|
| Number of subjects | 56 | 44 | |
| Gender | | | 0.47 |
| Male: n (%) | 28 (50) | 19 (42) | |
| Female: n (%) | 28 (50) | 25 (58) | |
| Age: Year (mean ± SD) | 39 ± 5 | 30±7 | 0.001 |
| Disease duration (mean year ± SD) | 16 ± 1.7 | | |
| PASI: n (mean ± SD) | 56 (16 ± 2) | | |
| PASI: (mild) | 23 (5 ± 0.5) | | |
| PASI: (moderate to server) | 33 (24 ± 2) | | |
| Medication | | | |
| Topical treatment: | 20 | | |
| Systemic treatment: | 36 | | |
| Adalimumab | 14 | | |
| Etanercept | 11 | | |
| Methotrexate | 4 | | |
| Ixekizumab | 3 | | |
| Secukinumab | 2 | | |
| Ustekinumab | 2 | | |

Data are presented as number, percentage (%) or mean ± standard deviation (SD). PASI, Psoriasis Area Severity Index.

blood buffy coat samples from 56 patients with psoriasis and 44 healthy controls. Relative quantitation of mtDNA-CN was calculated using the $2^{-\Delta\Delta ct}$ method and values are presented as mean ± SD. As shown in Fig 1, psoriasis patients exhibited a significantly lower mtDNA-CN compared with healthy controls (P = 0.04). The mean of mtDNA-CN was 93.6 ± 5.3 in the patients, and it was 205 ± 71 in the healthy controls. Since we have found a significant difference in age between psoriasis patients and healthy controls, we sought to determine the influence of age on mtDNA-CN. Therefore, the age of participants was included as a covariate in linear regression analysis in order to rule out its effect as a confounding factor. The results showed no significant association between age and mtDNA-CN ($r^2$ = 0.014, P = 0.47).

## mtDNA-CN is decreased with longer disease duration and disease severity

As we have shown a significant decrease in mtDNA-CN in psoriasis patients compared to controls, we next evaluated the association of mtDNA-CN with the disease duration as well as the disease severity. First, the patients were sub-divided into two groups based on disease duration which was recorded at the time from the first symptoms to blood collection as <10 years (n = 23) and >10 years (n = 33). As shown in Fig 2, the mtDNA-CN was significantly lower in patients with longer disease duration (P = 0.009). The mean of mtDNA-CN was 74 ± 4.3 in patients with disease duration >10 years, and it was 79 ± 8.3 in patients with disease duration <10 years. Next, the patients were sub-divided based on their PASI scores into two groups as mild (n = 23) and moderate to severe (n = 33). The results (Fig 3) showed that the mtDNA-CN was significantly lower in patients with higher PASI score (P = 0.017). The mean of mtDNA-CN was 72 ± 4.3 in patients with moderate-to-severe index, while it was 88.3 ± 6 in patients with mild index.

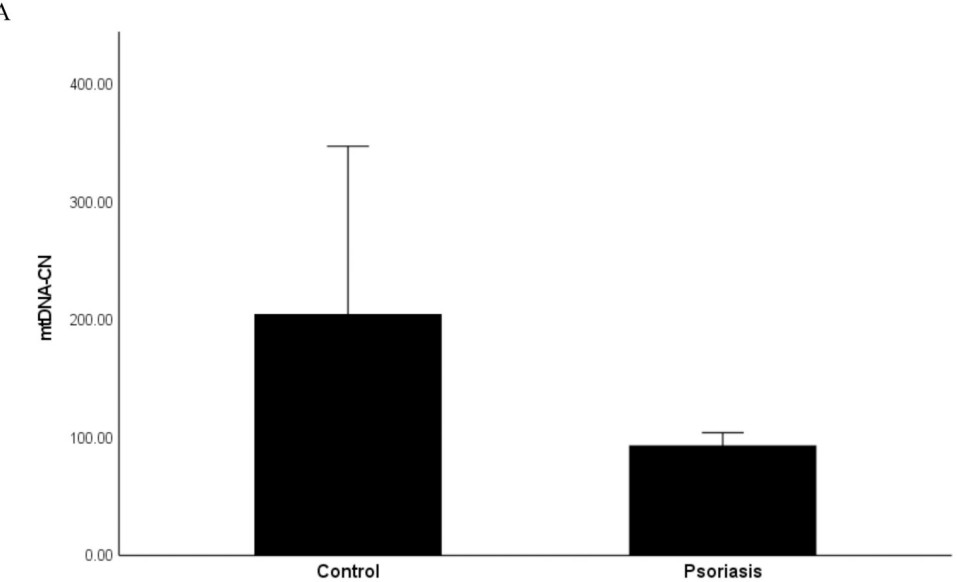

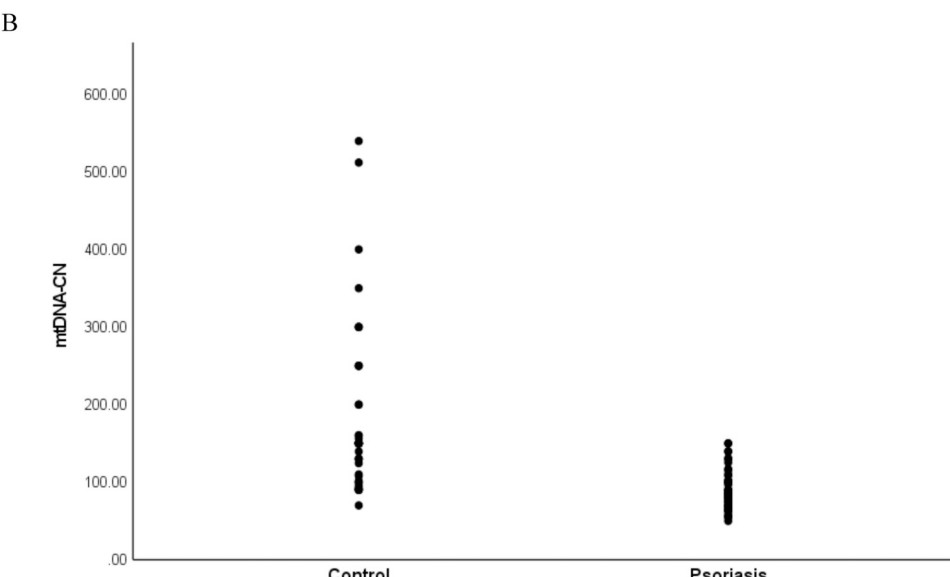

**Fig 1. mtDNA-CN in psoriasis patients and healthy controls.** The mtDNA-CN was quantified by real-time qPCR in peripheral blood buffy coat samples from patients with psoriasis (n = 56) and healthy controls (n = 44). A: Bar graph of data expressed as the mean ± standard deviation (SD). B: Dot plot of data.

### Correlation between mtDNA-CN and disease duration and severity

Spearman correlation analysis was performed to determine the association between mtDNA-CN and disease duration and severity as well as other clinical variables in the patient group (Table 2).

The results (Table 2) showed a significant negative correlation between mtDNA-CN and disease duration ($r = -0.36$, P = 0.006) as well as with the severity of the disease ($r = -0.41$, P = 0.003). No significant correlation was observed between mtDNA-CN and age ($r = 0.25$, P = 0.2).

A

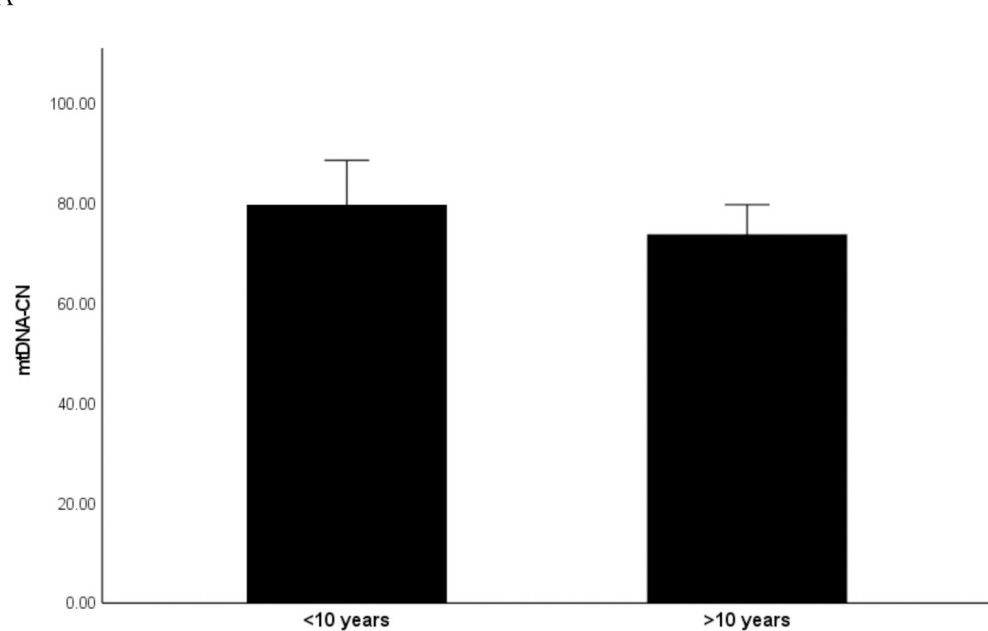

B

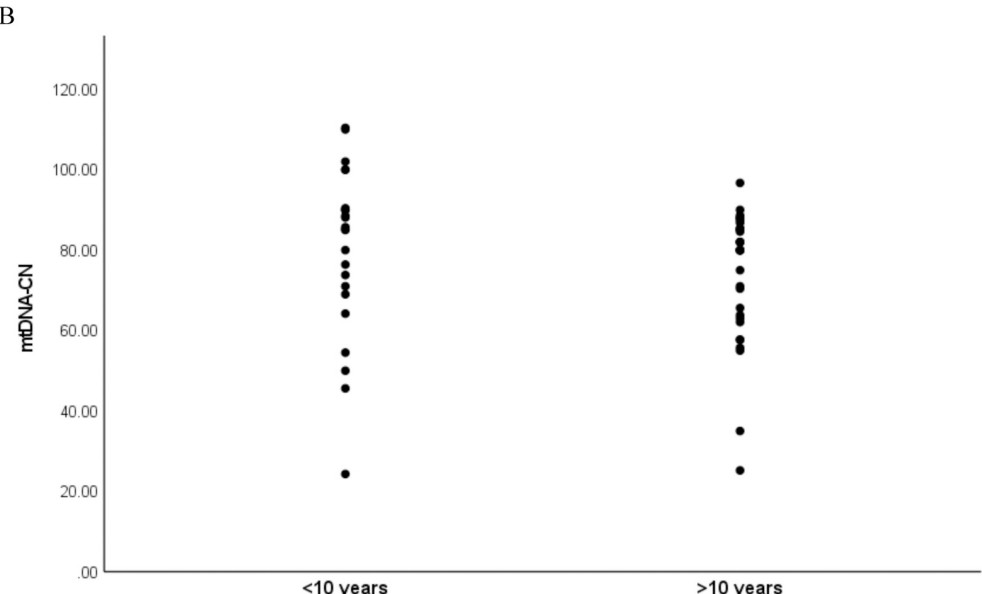

**Fig 2. Relationship of mtDNA-CN with disease duration.** The patients were sub-divided into two groups based on disease duration as <10 years (n = 23) and >10 years (n = 33). A: Bar graph of data expressed as the mean ± standard deviation (SD). B: Dot plot of data.

## Receiver operating characteristic analysis of mtDNA-CN

Receiver operating characteristic (ROC) analysis was performed to evaluate the diagnostic value of mtDNA-CN as a biomarker for psoriasis. The area under the curve (AUC) and 95% confidence interval (CI) were then calculated. The result (Fig 4) showed an AUC of 0.84 (95% CI: 0.694–0.977, P = 0.002) for mtDNA-CN to discriminate psoriasis patients from healthy controls.

A

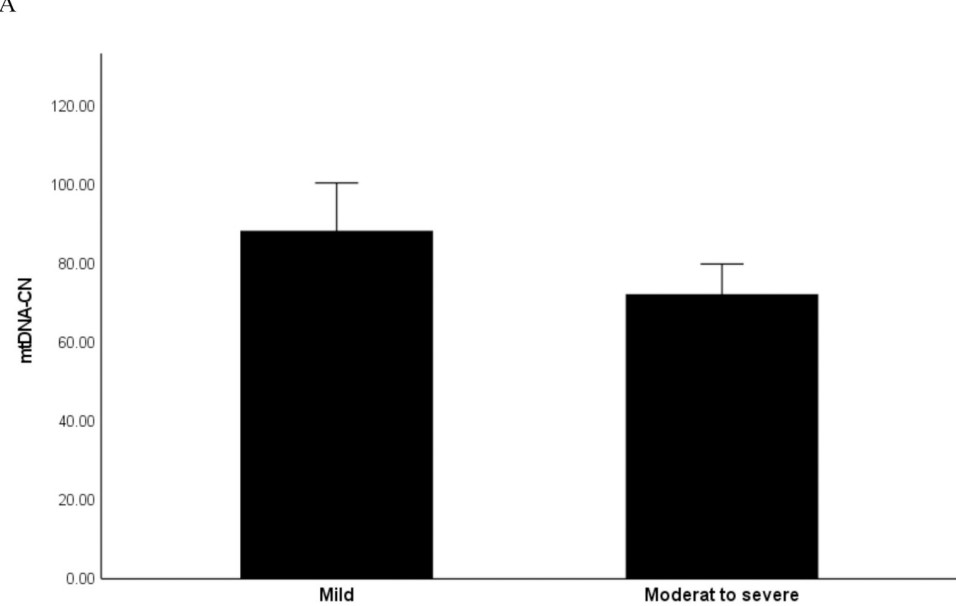

B

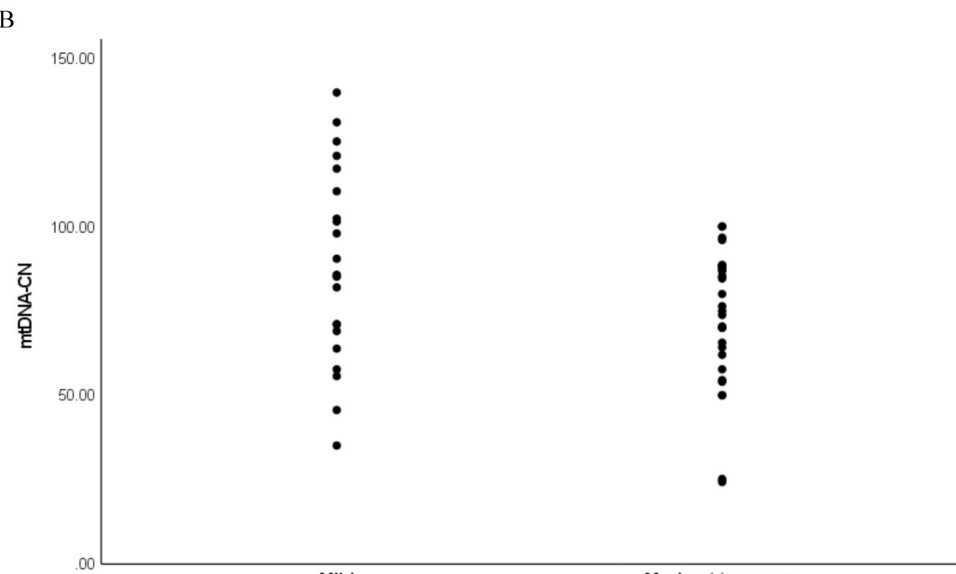

**Fig 3. Relationship of mtDNA-CN with the disease severity.** The patients were sub-divided based on PASI scores into two groups as mild (n = 23) and moderate-to-severe (n = 33). A: Bar graph of data expressed as the mean ± standard deviation (SD). B: Dot plot of data.

## Discussion

In the present study, we investigated changes in mtDNA-CN in the peripheral blood buffy coat of patients with psoriasis and healthy controls and examined the relationship of mtDNA-CN with the duration and severity of the disease as well as its potential use as a bio-marker for psoriasis. The results revealed that the mtDNA-CN was significantly lower in psoriasis patients compared to healthy controls. Stratification of patients by disease duration to

**Table 2. Correlation between mtDNA-CN number and clinical variables in psoriasis patients.**

| Parameters | r | P value |
|---|---|---|
| Disease duration | − 0.36 | 0.006 |
| Disease severity | − 0.41 | 0.003 |
| Age | 0.25 | 0.2 |

*r*, Spearman correlation.

under or over 10 years showed decreased mtDNA-CN in patients with longer disease duration. A trend towards decreased mtDNA-CN was also observed in patients with moderate to severe disease compared to patients with mild disease. These results suggest that low mtDNA content may be implicated in the pathogenicity of psoriasis. This suggestion was supported by the results of correlation analysis which revealed significant negative correlations of mtDNA-CN with the disease duration and disease severity. Results from this study also showed a significant ability of mtDNA-CN to differentiate patients from controls as indicated by the ROC analysis, suggesting a possible utility of mtDNA-CN as a biomarker for early detection of psoriasis.

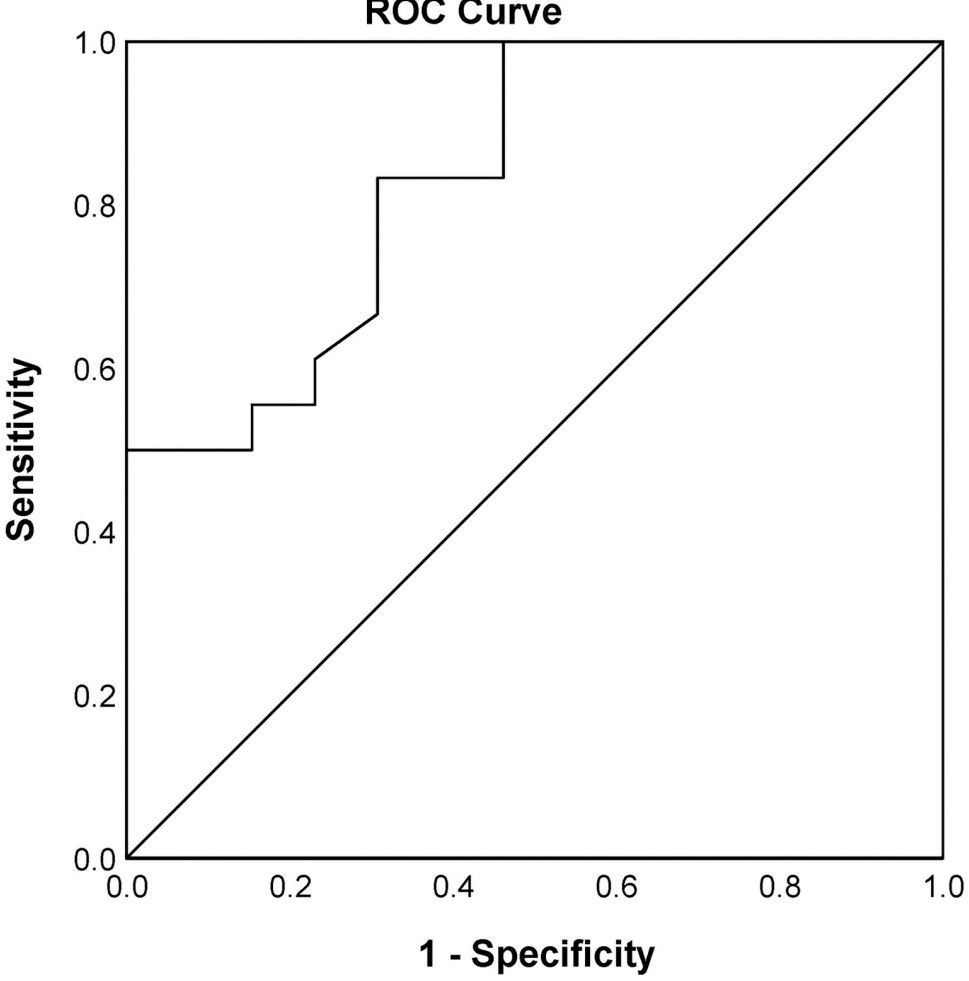

**Fig 4. ROC analysis of mtDNA-CN.** ROC curve was generated to evaluate the diagnostic value of mtDNA-CN as a biomarker for psoriasis. The AUC for discriminating psoriasis patients from healthy controls was 0.84 (95% CI: 0.694–0.977, P = 0.002).

Mitochondria is the main intracellular source of energy and the major site of reactive oxygen species (ROS) production. Human mitochondria have their own genome (mtDNA), which encodes essential subunit genes of the electron transport chain (ETC) complexes as well as tRNAs and rRNAs genes, all of which are important for normal mitochondrial function [13]. Compared to nuclear DNA (nDNA), the mtDNA is highly susceptible to oxidative stress and is a target of ROS attack due to its proximity to ROS-generating sites (respiratory chain), lack of a histone protection and limited DNA repair pathways [15]. Because mtDNA-CN reflects the abundance and function of mitochondria [14], increased oxidative stress and mtDNA damage will ultimately lead to impaired mitochondrial function with more ROS production [16–19].

Previous studies have revealed an essential role of mitochondrial defects in human aging and other pathological conditions including psoriasis [17–21]. Recent studies have shown that low mtDNA-CN is implicated in the pathogenesis of chronic diseases such as autoimmune diseases [29,30], cancer [31], type 2 diabetes [32], metabolic syndrome [33], type 2 diabetes, and the severity of coronary heart disease [35]. Although the exact mechanism(s) of decreased mtDNA-CN in diseases remain to be elucidated, oxidative stress and inflammation have been suggested as highly probable factors [21,22,24,37]. Specifically, increased oxidative stress is an important contributor to the pathogenesis of autoimmune diseases through enhancing inflammation, apoptotic cell death, and breaking down the immunological tolerance [38]. In psoriasis, a link between enhancement ROS production and decreased antioxidant defences has been suggested as a result of immunological and inflammatory mechanisms which are important in the etiopathogenesis of the disease [39]. Indeed, the mtDNA content in peripheral blood is associated with the overall level of oxidative stress [40] and suggested as a biomarker associated with oxidative stress and inflammation [41]. In most studies to date, mtDNA-CN has been measured in the buffy coat [29,33,34] or whole blood samples [30–32,35]. In our study, we observed lower mtDNA-CN in psoriasis patients compared to controls. We extracted DNA from peripheral blood buffy coat which contains all the white blood cells and platelets. Contrary to our results, a study by Therianou, et al [28] reported a significant increase in mtDNA content in serum from psoriatic patients compared to controls. Since mtDNA present in leukocytes and platelets, the use of serum can affect the accuracy of mtDNA-CN quantification [42], and this may explain the discrepancy in the two studies.

The reduction of mtDNA-CN in psoriasis patients observed in this study was correlated with the disease duration and disease severity, suggesting that low mtDNA-CN may be an early abnormality and associated with the disease progression. It is possible that low mtDNA-CN is a result of progressive impairment of oxidative phosphorylation and mitochondrial function, a speculation which needs to be investigated further. The link between mtDNA content and severity or outcome of diseases has been reported in several previous studies. For instance, a reduction in peripheral blood mtDNA-CN was correlated with the development of multiple sclerosis [30]. Lower peripheral mtDNA-CN was also associated with adverse clinical outcomes in peritoneal dialysis patients [43] as well as with the severity and inflammation in bipolar disorder [44]. Moreover, reduced mtDNA-CN has been shown to precede the development of type 2 diabetes [45] and observed in the early stages of neurodegenerative disorders including multiple sclerosis [30], Parkinson's disease [46] and Alzheimer's disease [47]. Whereas, low mtDNA-CN predict a poor outcome in hemodialysis patients with end-stage renal disease [48].

At present, there is no specific or diagnostic blood test for psoriasis. Previously evaluated markers are universal for inflammation and not specific to psoriasis [49]. Therefore, the development of non-invasive diagnostic tests or biomarkers for psoriasis is urgently needed. mtDNA-CN is an especially attractive biomarker because it can be non-invasively measured in

blood using a relatively easy and cheap method such as real-time PCR, in addition to the repeatability and reproducibility measurement of blood mtDNA-CN. Moreover, the observation that mtDNA-CN can differentiate patients from healthy controls, makes it a suitable biomarker for several diseases [29–32,50,51]. The present study also evaluated the diagnostic value of mtDNA-CN in psoriasis. mtDNA-CN was found to have a significant ability to differentiate psoriasis patients from healthy controls as indicated by the ROC analysis with a very good diagnostic value (AUC: 0.84, 95% CI: 0.694–0.977, P = 0.002). These findings suggest a potential use of peripheral blood mtDNA-CN as a biomarker for psoriasis.

The present study has certain limitations. First, the sample size was relatively small and future large-scale validation of the results is recommended, particularly the use of mtDNA-CN as a biomarker for psoriasis. Second, additional studies on the correlation of mtDNA-CN with inflammatory and oxidative stress markers can help in understanding the role of mtDNA content in the pathophysiologic mechanism(s) of psoriasis. Moreover, future studies to examine other blood indicators related to mitochondrial function and compare them with mtDNA are required to further clarify the influence of mtDAN on mitochondrial function. Finally, the possible changes of mtDNA-CN with different treatments should be evaluated in future studies to investigate the potential use of mtDNA-CN to monitor response to treatment.

## Conclusions

In this study we observed the mtDNA-CN in peripheral blood buffy coat was significantly reduced in patients with psoriasis. We also showed that decreased mtDNA-CN correlated with the disease duration and severity, suggesting that low mtDNA-CN may be an early abnormality in psoriasis and associated with the disease progression. Additionally, we showed the feasibility of mtDNA-CN as a non-invasive biomarker for psoriasis. Additional research is needed to assess whether these results are replicable in the future.

## Acknowledgments

We would like to thank Dr. Anantha Kethireddy and her colleagues at the Research Core Facility in the Health Sciences Center, Faculty of Medicine, Kuwait University for their technical support during this study.

## Author Contributions

**Conceptualization:** Materah Salem Alwehaidah, Ghada Al-Kafaji.

**Data curation:** Materah Salem Alwehaidah.

**Formal analysis:** Materah Salem Alwehaidah, Suad AlFadhli, Ghada Al-Kafaji.

**Investigation:** Materah Salem Alwehaidah, Ghada Al-Kafaji.

**Methodology:** Materah Salem Alwehaidah, Suad AlFadhli, Ghada Al-Kafaji.

**Project administration:** Materah Salem Alwehaidah.

**Resources:** Materah Salem Alwehaidah.

**Supervision:** Materah Salem Alwehaidah.

**Writing – original draft:** Materah Salem Alwehaidah.

**Writing – review & editing:** Materah Salem Alwehaidah, Ghada Al-Kafaji.

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
