## [Decision Letter · Decision Letter 0]

21 Apr 2022

PONE-D-22-07315Leukocyte mitochondrial DNA copy number is a potential non-invasive biomarker for psoriasisPLOS ONE

Dear Dr. salem,

Thank you for submitting your manuscript to PLOS ONE. After careful consideration, we feel that it has merit but does not fully meet PLOS ONE’s publication criteria as it currently stands. Therefore, we invite you to submit a revised version of the manuscript that addresses the points raised during the review process.

We look forward to receiving your revised manuscript.

Kind regards,

Tomoyoshi Komiyama, Ph.D

Academic Editor

PLOS ONE

Journal Requirements:

1 Please ensure that your manuscript meets PLOS ONE's style requirements, including those for file naming. The PLOS ONE style templates can be found at 

Additional Editor Comments:

Dear Authors,

The focus of your study was to investigate whether the mtDNA-CN is related to psoriasis and can serve as a disease biomarker.　In the results, the study demonstrated low mtDNA-CN in psoriasis patients compared to controls for the first time. In addition, it found that the reduction in mtDNA-CN was correlated with the disease duration and severity, suggesting that lower mtDNA-CN may be an early abnormality in psoriasis, and is also associated with the disease progression.

If these results become clear, I think your research is important for the future of clinical treatment of patients diagnosed with psoriasis, research on the causes of psoriasis, and clinical researchers who aim to better understand psoriasis characteristics during unconstrained daily activities.

However, I think that it is necessary to strengthen the reliability of the result by adding as much information as possible.

Based on two reviewer’s comments, I believe some major revision may be needed.

I have attached the comments from these reviewers about the manuscript that you might consider.

I think that the points suggested by these reviews will be helpful in the revision.

Tomoyoshi Komiyama

Reviewers' comments:

Reviewer's Responses to Questions

**Comments to the Author**

1. Is the manuscript technically sound, and do the data support the conclusions?

Reviewer #1: Yes

Reviewer #2: Partly

2. Has the statistical analysis been performed appropriately and rigorously? 

Reviewer #1: No

Reviewer #2: No

3. Have the authors made all data underlying the findings in their manuscript fully available?

Reviewer #1: Yes

Reviewer #2: No

4. Is the manuscript presented in an intelligible fashion and written in standard English?

Reviewer #1: Yes

Reviewer #2: Yes

5. Review Comments to the Author

Reviewer #1: In this study, these authors have investigated that mtDNA-CN is a potential biomarker for psoriasis. This is an important finding in the field of psoriasis. The manuscript is compact and well written. Below are comments aimed at improving this important study even further:

1. All data present bar graph. I recommend to refine bar graph with overlapping dots plot or dot plot (Figure 1-3).

2. These authors described “the first to measure mtDNA-CN in psoriasis patients” in Abstract. I believe that this is true, however this sentence is prone to trouble for Journal (Plos One). I recommend to omit “the first”.

3. I recommend to refine detailed statistical significance in Abstract (page 2 line 12 and 14).

Reviewer #2: The manuscript entitled " Leukocyte mitochondrial DNA copy number is a potential non-invasive biomarker for psoriasis " describes a clinical study aimed to investigate the role of mtDMA in psoriasis. Although the experimental design and methods overall look appropriate, the manuscript requires significant revision.

Q1: Before treatment, there was a significant difference in age between the psoriasis group and the normal group. How to exclude the influence of age in this study?

Q2: This study showed a decrease in mtDNA in the serum of patients with psoriasis and investigated the relationship between mtDNA and the duration and severity of the disease, as well as its potential use as a biomarker for psoriasis. However, the relationship between mtDNA and mitochondrial function has not been directly proved here. It is suggested that the author add other blood examination indicator related to mitochondrial function and compare them with mtDNA to further clarify the influence of mtDAN on mitochondrial function.

Q3: Psoriasis is a chronic immune-mediated inflammatory skin disease, which is closely related to the function of mitochondria. The authors note: “Oxidative damage to mtDNA can result in mutations and alterations in mtDNA replication and/or transcription efficiency and mtDNA-CN, which may subsequently lead to a decline in mitochondrial functionwith more ROS production”. Mitochondria are the main sites of ROS production, and high concentration of ROS can promote oxidative stress response, and the increase of oxidative stress and inflammation will further induce mitochondrial damage, leading to cell apoptosis. However, the author did not detect ROS and inflammation indicators, so it is suggested to add the content of this part to clarify the effects of mtDNA damage on ROS, inflammation and apoptosis, and whether mtDNA changes affect psoriasis by regulating the above pathways?

Q4: Please add the code/number of ethics in the manuscript.

Q5: Statistical analyses require clarification. Were the data normally distributed? Please indicate in the chart notes which data conform to normal distribution and test method used.

Q6: Please provide a statement of interest.

6. PLOS authors have the option to publish the peer review history of their article (what does this mean?). If published, this will include your full peer review and any attached files.

Reviewer #1: No

Reviewer #2: No

---

## [Author Response · Author response to Decision Letter 0]

21 May 2022

Reviewer #1: In this study, these authors have investigated that mtDNA-CN is a potential biomarker for psoriasis. This is an important finding in the field of psoriasis. The manuscript is compact and well written. Below are comments aimed at improving this important study even further:

1. All data present bar graph. I recommend to refine bar graph with overlapping dots plot or dot plot (Figure 1-3).

Our response

Dots plot along with the bar graphs for Figure 1-3 are added in the revised manuscript (yellow highlight).

2. These authors described “the first to measure mtDNA-CN in psoriasis patients” in Abstract. I believe that this is true, however this sentence is prone to trouble for Journal (Plos One). I recommend to omit “the first”.

Our response

This sentence is deleted in the revised manuscript.

3. I recommend to refine detailed statistical significance in Abstract (page 2 line 12 and 14).

Our response

Detailed statistical significance in Abstract (page 2 line 12 and 14) is improved in the revised manuscript.

Reviewer #2: The manuscript entitled " Leukocyte mitochondrial DNA copy number is a potential non-invasive biomarker for psoriasis " describes a clinical study aimed to investigate the role of mtDMA in psoriasis. Although the experimental design and methods overall look appropriate, the manuscript requires significant revision.

Q1: Before treatment, there was a significant difference in age between the psoriasis group and the normal group. How to exclude the influence of age in this study? 

Our response

Age of participants was included as covariate in linear regression in order to rule out confounding effect. The results showed no significant association between age and mtDNA-CN (r2 = 0.014, P=0.47). This statement is included in the results of the revised manuscript (yellow highlight). 

Q2: This study showed a decrease in mtDNA in the serum of patients with psoriasis and investigated the relationship between mtDNA and the duration and severity of the disease, as well as its potential use as a biomarker for psoriasis. However, the relationship between mtDNA and mitochondrial function has not been directly proved here. It is suggested that the author add other blood examination indicator related to mitochondrial function and compare them with mtDNA to further clarify the influence of mtDAN on mitochondrial function. 

Our response

We agree with this valuable comment of the reviewer. However, previous reports have clearly shown impaired mtDNA and mitochondrial function in psoriasis (as mentioned in the introduction). Based on these previous studies, the aim of our study was to investigate changes in mtDN copy number (mtDNA-CN) in blood of psoriasis patients and the potential use of mtDNA-CN as a biomarker for psoriasis. 

The suggestion of the reviewer to add other blood examination indicators related to mitochondrial function and compare them with mtDNA to further clarify the influence of mtDNA on mitochondrial function will be added as a limitation in our study, in the conclusion section of the revised manuscript (yellow highlight).

Q3: Psoriasis is a chronic immune-mediated inflammatory skin disease, which is closely related to the function of mitochondria. The authors note: “Oxidative damage to mtDNA can result in mutations and alterations in mtDNA replication and/or transcription efficiency and mtDNA-CN, which may subsequently lead to a decline in mitochondrial function with more ROS production”. Mitochondria are the main sites of ROS production, and high concentration of ROS can promote oxidative stress response, and the increase of oxidative stress and inflammation will further induce mitochondrial damage, leading to cell apoptosis. However, the author did not detect ROS and inflammation indicators, so it is suggested to add the content of this part to clarify the effects of mtDNA damage on ROS, inflammation and apoptosis, and whether mtDNA changes affect psoriasis by regulating the above pathways?

Our response

Since previous studies have shown a clear relationship between mtDNA-CN and increased ROS and inflammation, we focused on our main aim which is investigating the potential use of mtDNA-CN as a blood-based biomarker for psoriasis. Nevertheless, we mentioned this suggestion as a future studies in the conclusion section of the original manuscript as follows: “additional studies on the correlation of mtDNA-CN with inflammatory and oxidative stress markers can help in understanding the role of mtDNA content in the pathophysiologic mechanism(s) of psoriasis”.

Q4: Please add the code/number of ethics in the manuscript. 

Our response

The number of ethics is added in the method section of the revised manuscript (yellow highlight): “No. 2016/496”. 

Q5: Statistical analyses require clarification. Were the data normally distributed? Please indicate in the chart notes which data conform to normal distribution and test method used.

Our response

The normal distribution of the data of the study group was assessed by the Kolmogorov-Smirnov. This is already mentioned in the statistical analysis section of the original manuscript.

In the revised manuscript we also added the test results (P=0.2) confirming the normal distribution of data (yellow highlight). 

Q6: Please provide a statement of interest.

Our response

The following section is added in the revised manuscript (yellow highlight)

“Competing interests: The authors declare that they have no competing interests”.

---

## [Decision Letter · Decision Letter 1]

10 Jun 2022

PONE-D-22-07315R1Leukocyte mitochondrial DNA copy number is a potential non-invasive biomarker for psoriasisPLOS ONE

Dear Dr. salem,

Thank you for submitting your manuscript to PLOS ONE. After careful consideration, we feel that it has merit but does not fully meet PLOS ONE’s publication criteria as it currently stands. Therefore, we invite you to submit a revised version of the manuscript that addresses the points raised during the review process.

We look forward to receiving your revised manuscript.

Kind regards,

Tomoyoshi Komiyama, Ph.D

Academic Editor

PLOS ONE

Journal Requirements:

Additional Editor Comments:

Dear authors

Your study found that reduction in mtDNA-CN was correlated with disease duration and severity, suggesting that lower mtDNA-CN may be an early abnormality in psoriasis and is associated with disease progression.

I thought your study of the genetic marker for psoriasis was very interesting on a potential. Also, I think that the authors responded appropriately to the comments from the two reviewers. However, please correct some further points as noted below.

1.The purposes and results are slightly inconsistent across the abstract, introduction, discussion, and conclusion sections, please ensure they are the same throughout the manuscript so that they are easy to understand.

2. Please combine the known mitochondrial information that forms the basis of the discussion section with the introduction section.

3. Make the conclusion a little clearer. For limitation, please make a separate limitation section or add it to the discussion section as this needs to be described separately to the conclusion.

Reviewers' comments:

Reviewer's Responses to Questions

**Comments to the Author**

1. If the authors have adequately addressed your comments raised in a previous round of review and you feel that this manuscript is now acceptable for publication, you may indicate that here to bypass the “Comments to the Author” section, enter your conflict of interest statement in the “Confidential to Editor” section, and submit your "Accept" recommendation.

Reviewer #1: All comments have been addressed

Reviewer #2: (No Response)

2. Is the manuscript technically sound, and do the data support the conclusions?

Reviewer #1: Yes

Reviewer #2: Partly

3. Has the statistical analysis been performed appropriately and rigorously? 

Reviewer #1: Yes

Reviewer #2: I Don't Know

4. Have the authors made all data underlying the findings in their manuscript fully available?

Reviewer #1: Yes

Reviewer #2: Yes

5. Is the manuscript presented in an intelligible fashion and written in standard English?

Reviewer #1: Yes

Reviewer #2: Yes

6. Review Comments to the Author

Reviewer #1: In this study, these authors have investigated that mtDNA-CN is a potential biomarker for psoriasis.

This is an important finding in the field of psoriasis.

The manuscript is compact and well written.

In addition, I found that the manuscript has been revised well.

Reviewer #2: (No Response)

7. PLOS authors have the option to publish the peer review history of their article (what does this mean?). If published, this will include your full peer review and any attached files.

Reviewer #1: **Yes: **Kazuhito Gotoh

Reviewer #2: No

---

## [Author Response · Author response to Decision Letter 1]

14 Jun 2022

Dear Editor,

Thanks for your valuable comments. Please find below our response:

1. The purposes and results are slightly inconsistent across the abstract, introduction, discussion, and conclusion sections, please ensure they are the same throughout the manuscript so that they are easy to understand.

Response 

The suggested changes have been done all over the manuscript (yellow highlights).

2. Please combine the known mitochondrial information that forms the basis of the discussion section with the introduction section.

Response

The suggested changes have been done all over the manuscript (yellow highlights). Please note that because of these changes, some redundant references have been removed and the order of the references has been changed (yellow highlights). 

3. Make the conclusion a little clearer. For limitation, please make a separate limitation section or add it to the discussion section as this needs to be described separately to the conclusion.

Response

The conclusion has been made clearer and the limitations section has been moved to the discussion (yellow highlights).

---

## [Editor Report · Decision Letter 2]

16 Jun 2022

Leukocyte mitochondrial DNA copy number is a potential non-invasive biomarker for psoriasis

PONE-D-22-07315R2

Dear Dr. salem,

We’re pleased to inform you that your manuscript has been judged scientifically suitable for publication and will be formally accepted for publication once it meets all outstanding technical requirements.

Kind regards,

Tomoyoshi Komiyama, Ph.D

Academic Editor

PLOS ONE

Additional Editor Comments (optional):

Dear authors,

Thank you for submitting your revised manuscript.

I think it was much easier to understand than the original manuscript.

The authors revised the sentences according to reviewer’s comments.

I am satisfied with the responses and the edits, I am happy to accept this manuscript.

Therefore, I have no further questions.

Tomoyoshi Komiyama
---

## [Editor Report · Acceptance letter]

20 Jun 2022

PONE-D-22-07315R2 

Leukocyte mitochondrial DNA copy number is a potential non-invasive biomarker for psoriasis 

Dear Dr. Alwehaidah:

I'm pleased to inform you that your manuscript has been deemed suitable for publication in PLOS ONE. Congratulations! Your manuscript is now with our production department. 

Kind regards, 

on behalf of

Dr. Tomoyoshi Komiyama 

Academic Editor

PLOS ONE